# Trends in Quantification of HbA1c Using Electrochemical and Point-of-Care Analyzers

**DOI:** 10.3390/s23041901

**Published:** 2023-02-08

**Authors:** Pavan Kumar Mandali, Amrish Prabakaran, Kasthuri Annadurai, Uma Maheswari Krishnan

**Affiliations:** 1Centre for Nanotechnology& Advanced Biomaterials, SASTRA Deemed University, Thanjavur 613 401, India; 2School of Chemical and Biotechnology, SASTRA Deemed University, Thanjavur 613 401, India; 3School of Arts, Sciences, Humanities & Education, SASTRA Deemed University, Thanjavur 613 401, India

**Keywords:** glycated hemoglobin, HbA1c, electrochemical sensors, biosensors, PoC devices

## Abstract

Glycated hemoglobin (HbA1c), one of the many variants of hemoglobin (Hb), serves as a standard biomarker of diabetes, as it assesses the long-term glycemic status of the individual for the previous 90–120 days. HbA1c levels in blood are stable and do not fluctuate when compared to the random blood glucose levels. The normal level of HbA1c is 4–6.0%, while concentrations > 6.5% denote diabetes. Conventionally, HbA1c is measured using techniques such as chromatography, spectroscopy, immunoassays, capillary electrophoresis, fluorometry, etc., that are time-consuming, expensive, and involve complex procedures and skilled personnel. These limitations have spurred development of sensors incorporating nanostructured materials that can aid in specific and accurate quantification of HbA1c. Various chemical and biological sensing elements with and without nanoparticle interfaces have been explored for HbA1c detection. Attempts are underway to improve the detection speed, increase accuracy, and reduce sample volumes and detection costs through different combinations of nanomaterials, interfaces, capture elements, and measurement techniques. This review elaborates on the recent advances in the realm of electrochemical detection for HbA1c detection. It also discusses the emerging trends and challenges in the fabrication of effective, accurate, and cost-effective point-of-care (PoC) devices for HbA1c and the potential way forward.

## 1. Introduction

Diabetes is a metabolic disorder manifested through elevated blood glucose levels and is commonly associated with complications such as microvascular and cardiovascular disorders, vision impairment (retinopathy), kidney dysfunction (nephropathy), peripheral nerve disorders (neuropathy), etc. [1,2,3,4,5]. The global prevalence of diabetes nearly quadrupled from 1980 to 2014, and the current estimates by the International Diabetes Federation suggest over 537 million diabetic individuals worldwide [6]. These numbers are expected to further increase to 643 million by the year 2030 [6,7]. Unhealthy lifestyle and diet, as well as genetic causes, have been mainly implicated in the onset of diabetes. Late diagnosis of diabetes can be life-threatening. Early detection of diabetes can lead to lesser organ damage by introducing non-pharmacological interventions in the form of lifestyle modifications.

Random blood glucose is a direct measure of glucose levels in the blood and has been traditionally used to obtain information on the glycemic status of an individual. However, glucose levels are prone to fluctuations owing to numerous factors, such as diet and physical and emotional stress [8]. Hence, there is a need for a more stable and reliable marker for diabetes. Glycated hemoglobin (HbA1c), a stable compound belonging to the class of advanced glycation end products (AGE), is formed as a result of the non-enzymatic addition of various sugars to hemoglobin (Hb). It comprises about 5% of the total Hb in human blood. HbA1c is further classified based on the sugar adducts attached to Hb. The HbA1a1 form has fructose-1,6-diphosphate as the substituent, HbA1a2 has glucose-6-phosphate, HbA1b has pyruvate, while HbA1c has glucose as the substituent. Among these, HbA1c is the most abundant form, formed by a non-enzymatic reaction, and reflects the glycemic condition of the individual over 2 to 3 months [9,10]. Hence, it serves as a stable biomarker for diabetes in blood, unlike glucose [9,11].

HbA1c is formed in a two-step non-enzymatic reaction, where glucose adds to the valine residue in the β-chain of Hb, forming a Schiff’s base, which is then rearranged yielding HbA1c, an Amadori product [10,12] (Figure 1). The extent of glycation may vary, with the number of glycated sites up to 15 being reported [13]. Clinically, HbA1c levels are usually expressed as a percentage of total Hb to account for the variations in the Hb levels among individuals. To facilitate user-friendly interaction, a linear relationship between HbA1c and equivalent average glucose (AG) levels was proposed by the A1c-Derived Average Glucose (ADAG) Study Group using the following relation [14,15]:$$\mathrm{eAG}\;\left(\frac{\mathrm{mg}}{\mathrm{dL}}\;\right)=\left(28.7*\mathrm{HbA}1c\%\right)-46.7$$

The normal range of HbA1c in healthy individuals lies between 4 and 6.0%, while in severe hyperglycemic conditions, HbA1c values can go up to 20%. In general, individuals with HbA1c values in the range of 5.7–6.4% are classified in the prediabetic stage, while the individuals having HbA1c > 6.5% are diabetic [16,17]. During the recent pandemic, it was shown that diabetic individuals with HbA1c levels > 8.66% had a higher risk of life-threatening complications when infected with the SARS-CoV2 coronavirus, thereby necessitating the need for a robust and rapid detection strategy for HbA1c to screen the large numbers of infected individuals [18,19,20,21,22].

## 2. Sensors for Quantification of HbA1c

The emerging field of sensors offers a wide versatile platform for the detection of various analytes, which can be tailored based on the desired analyte, employing appropriate detection elements. Sensors comprise a sensing element that responds to the analyte and a transducer that produces the readable output. If the sensing element involves a chemical molecule, the sensor is classified as a chemical sensor, while those that employ biomolecules such as enzymes, antibodies, aptamers, oligonucleotides, etc., are classified as biosensors [23]. While chemical sensors are cost-effective and stable, they do not possess the specificity and sensitivity of biosensors [23]. Sensors are also classified based on the transduction element as optical, electrochemical, electrical, calorimetric, and piezoelectric sensors. Among the various transduction methods, electrochemical sensors are widely employed for analytical applications due to the simple operation, rapid response, cost-effectiveness, portability, and ease of fabrication [24]. Though there are several reports of HbA1c quantification using optical [25], photoacoustic [26], electrochemical piezoelectric [27], chemiluminescence [28], photoelectric [29], magnetic immunoassay [30], etc., majority of the reports have focused on electrochemical methods of transduction owing to their merits. This review focuses only on the progress in electrochemical strategies for HbA1c measurement. The electrochemical sensors can be further classified as potentiometric sensors, that measure the charge potential accumulated in the working electrode, amperometric sensors, that measure the change in current due to reduction and oxidation of electroactive species in the system at a constant potential, voltametric sensors, that measure the current change at a defined potential window, and impedimetric sensors, that detect changes in the capacitance and charge conductance at the electrode/electrolyte interface [31].

In order to improve the sensitivity, response times, and detection range of the electrochemical sensors, the earlier generations of electrochemical sensors employed an electron transfer agent or a redox mediator, while the current generation of sensors employ direct electron transfer on an electrode surface that is modified with nano-dimensional particles. These nanointerfaces offer amplified electro-catalytical, electron transfer, and conductivity owing to their structural characteristics. Nanostructures of various sizes, shapes, and chemical nature have been explored as interface materials. Further, these nanostructured interfaces also serve as immobilization matrices with a high surface area for retaining biological capture elements such as enzymes and antibodies, thereby augmenting the specificity, sensitivity, and detection range of the biosensor.

Electrochemical measurements of HbA1c have been carried out by both chemical and biosensors. In the chemical method, the conventional cis-diol interaction between boronate and the sugar moiety of HbA1c is exploited for the detection of HbA1c, while biosensors quantify the redox changes or current flow variations owing to interactions between HbA1c and the biological recognition elements, such as antibodies or enzymes. Additionally, HbA1c has also been quantified in an indirect method which involves the detection of fructosyl-valine (FV), a small peptide generated by digesting HbA1c by oxidation using the enzyme fructosyl amino acid oxidase (FAO), resulting in the generation of H_2_O_2_, which is measured electrochemically [32]. The present review includes science citation indexed articles published from the year 2000 to present in Science Direct, PubMed, and Google Scholar on electrochemical sensors for HbA1c. Sensors fabricated with either chemical or biological sensing elements have been considered for the review. Nano-interfaced and point-of-care sensors for HbA1c have been emphasized upon. Similar interfaces and electrode modification strategies have been excluded for minimizing redundancy.

### 2.1. Electrochemical Sensors for Direct HbA1c Detection Using Boronate Chemistry

For HbA1c sensing, the electrochemical sensors work on the principle of boronic acid interactions with the diol groups in HbA1c, which are absent in normal Hb, thereby conferring specificity for the detection (Figure 2). The frequently used boronic acid derivatives are ferrocene boronic acid (FcBA), aminophenyl boronic acid (APBA), formylphenyl boronic acid (FPBA), thiophene-3-boronic acid (T3BA), etc. These boronic acid derivatives have been sometimes used in tandem with metallic nanoparticles for increased conductivity and sensing performance (Table 1).

In an earlier work, an amperometric sensor employing FcBA as the capture element was used for HbA1c detection. The pyrolytic graphite working electrode (PGE) modified with zirconium dioxide (ZrO_2_) nanoparticles was used to immobilize Hb or HbA1c from the samples. The working electrode was then incubated in FcBA solution and the amount of HbA1c was measured through the electrochemical response of FcBA using square wave voltammetry (SWV). The electrochemical response of heme was used to quantify total hemoglobin (tHb) in the PGE without the addition of FcBA. The sensor exhibited a linear response between 6.8 and 14% of HbA1c. The sensor performance was validated using whole blood samples from human volunteers and compared with the conventional HPLC method. A variation ranging from −10.7 to 31% was observed [33]. The limitations of this system are the tedious and long sample analysis time, the narrow detection range, as well as greater deviations from the values reported by the conventional methods.

Screen-printed electrodes have garnered much attention in the clinical diagnosis domain owing to their cost-effectiveness and disposability. Liu et al. [34] designed a novel HbA1c sensor using a screen-printed reticulated vitreous carbon (RVC) electrode. For fabrication of the sensor, RVC foam was compressed to form a working electrode of diameter 3 mm and length 5 mm, which was further incubated in HNO_3_ and H_2_SO_4_ solution for 1 h at 85 °C. Then, APBA was immobilized on the working electrode using an ethyl (dimethyl amino propyl) carbodiimide–N-hydroxysuccinimide (EDC-NHS) coupling reaction. Finally, the electrode was immersed in glycine solution to block the unreacted carboxylic ends of RVC. This modified RVC was set into a cylindrical polyether ether ketone (PEEK) to form the flow cell. The electrochemical properties of the electrode were studied in the flow injection assay system. The results showed that the sensor exhibited a detection range of 0.2–12.0 mg/mL HbA1c with a limit of detection (LoD) of 89 μg/mL at a low potential of +0.2 V. The electrode was also stable and lost only 4.4% of the signal intensity after 100 cycles of use.

Another disposable APBA-based sensor fabricated on screen-printed carbon electrodes (SPCE) comprised of a layer containing electrodeposited gold nanoparticles (AuNPs) that was dipped in a 1 mM TTBA (2,2′:5′,5″-terthiophene-3′-p-benzoic acid) solution. The TTBA was then electro-polymerized by potential cycling twice from 0.0 to +1.4 V (Ag/AgCl) in 0.1 M phosphate-buffered saline (PBS) (pH 7.4) at a scan rate of 0.1 V/s, followed by immobilization of APBA using EDC-NHS chemistry. The final working electrode (APBA-pTTBA/AuNPs/SPCE) when subjected to electrochemical studies demonstrated the specificity and stability of the HbA1c owing to the highly stable conducting polymer layer. The sensor had a LoD of the sensor of 0.052% and showed good correlation with conventional HPLC in the HbA1c values measured for diabetic and normal patients [35]. In a similar strategy, 4-mercaptophenylboronic acid (4-MPBA) was immobilized by drop-casting on a SPCE working electrode modified with an electrodeposited layer of gold nanoflowers (AuNF). The resulting 4-MPBA/AuNF/SPCE was characterized and then employed for the quantification of HbA1c and exhibited a linear range between 5 μg/mL and 1 mg/mL of HbA1c [36].

In a different approach, a boronic acid-modified gold electrode was fabricated by dipping the electrode in a DMSO (dimethyl sulfoxide) solution of 3,3′-Dithio-bis-propionic acid N-hydroxysuccinimide ester (DTSP) to form an amine-reactive self-assembled monolayer (SAM). The electrode was then incubated in a 0.5% (*w*/*w*) ethanolic solution of poly(amidoamine) generation 4 (PAMAM) dendrimer and then in 1 mM of 4-formyl-phenylboronic acid (FPBA). The sensing strategy employs a unique “backfilling” assay scheme (Figure 3), where after addition of the sample containing HbA1c, periodate-treated activated glucose oxidase (GOx) was made to react with the unreacted amine groups on the PAMAM-FPBA electrode, which produced the electrochemical response. The signal intensity decreased with increased concentrations of HbA1c due to steric interference of HbA1c on the catalytic efficiency of GOx, thereby providing a stable, quantitative measurement of HbA1c in the sample. The sensor exhibited a detection range of 2.5–15%, which was well within the clinical levels for diagnosis. However, the sensor requires further validation with clinical samples to establish its diagnostic potential [37]. The same group have also reported an interface comprising a conjugate of cysteamine and FPBA that were allowed to react for 4 h at 60 °C under stirring conditions to obtain the Cys-FPBA_2_ conjugate. The gold electrodes were dipped in Cys-FPBA_2_ for the formation of a boronate-modified SAM on the working electrode. The electrochemical response was obtained using the backfilling method with GOx. The Hb was separated from whole blood by zinc-induced precipitation. The sensor exhibited a detection range of 4.5–15%. The sensor performance was evaluated with a small sample size of human samples. Though promising, the sensor preparation involves a larger number of steps, which increased the detection time [38].

Poly(3-aminophenyl boronic acid) (PAPBA) nanoparticles obtained by polymerization of APBA have been employed as a thin-film nanointerface material over SPCE for detection of HbA1c [39]. The sensor exhibited a linear range between 0.975 and 156 μM of HbA1c when measured using differential pulse voltammetry and was stable when used in the presence of interferents such as dopamine, ascorbic acid, glucose, uric acid, and bovine serum albumin. However, the sensor did not account for potential errors arising due to variations in the total hemoglobin levels.

Recently, a novel interface of anthraquinone boronic acid (AQBA) that was formed by electrochemical oxidation of 2-anthraceneboronic acid (ANBA) on pre-anodized SPCE, designated as AQBA/SPCE*, was employed for HbA1c quantification [40]. The sensor performance remained unaffected in the presence of interferents such as ascorbic acid, uric acid, dopamine, glucose, fructose, mannose, and hemoglobin. The sensor detected HbA1c in the linear range of 31.2–500 nM. An AQBA-free electrode was used for detecting Hb. The sensor exhibited good correlation with the results obtained from the conventional HPLC method for HbA1c in patient samples.

Several studies have employed electrochemical impedance spectroscopic (EIS) measurements to measure the rate of charge transfer between the electrode and the redox probe, which is influenced the blocking effect caused by binding of HbA1c with the recognition element (boronic acid derivatives). An earlier study had employed screen-printed platinum electrodes, surface-coated with a fibrous mesh of eggshell membrane, incorporating APBA through glutaraldehyde crosslinking. The hemolyzed sample was immobilized on the electrode surface and changes in the impedance were used for the label-free quantification of HbA1c between 2.3 and 14%. A ferro-ferricyanide couple was employed as the redox mediator in this study [41]. The study employed centrifugation to remove other glycated moieties that could potentially interfere with the analysis. While this is beneficial for the accuracy of measurement, it also makes the process time-consuming and tedious. In a similar strategy, interdigitated gold electrodes modified with a self-assembled monolayer of cysteamine covalently linked to APBA through glutaraldehyde were employed for quantification of HbA1c using impedance measurements. Hemolyzed whole blood was immobilized on the electrode surface for the measurements. The sensor exhibited a linear range of 0.1–8.36% of HbA1c with a LoD of 0.024% [42]. However, the clinical utility of this system could be improved with an increase in the upper threshold of detection. Use of gold electrodes also reduces the cost-effectiveness of the measurement. In a similar study employing EIS for quantification of HbA1c, gold electrodes modified with a self-assembled monolayer of T3BA were employed to achieve a detection range between 0.08 and 8.4% of HbA1c [43]. The sample preparation in this case involved removal of glycated albumin by centrifugation to avoid interferences from other glycated species in the sample. However, this additional step will not only increase the total analysis time, but also restrict the portability of the system.

Graphene oxide (GO) modified with APBA has also been employed for the impedimetric-based quantification of HbA1c. In this work, 1-Ethyl-3-(3-dimethylaminopropyl) carbodiimide (EDC), a coupling agent, was employed to facilitate the formation of an amide bond between the carboxylic group of GO and the amine group of APBA. The synthesized GO-APBA was drop-casted onto a clean glassy carbon electrode and employed for further electrochemical studies. The sensor could detect HbA1c in the range of 8% [44]. However, the study did not specify the limit of detection, linear range of detection of HbA1c, or the effect of interferents on the sensor performance. Therefore, the true clinical effectiveness of this system could not be predicted.

Hsieh et al. developed a ring-shaped interdigitated electrode (RSIDE) that works on impedance spectroscopy for the quantification of HbA1c. Here, the gold electrode surface was modified with a SAM of T3BA. The sensor design was evaluated on a computational platform using COSMOL simulations, which was then translated to a biosensor chip for quantifying HbA1c. The sensor measured HbA1c in the range of 1–100 mg/L. The cheap, label-free, and easy micro-electro-mechanical system (MEMS) fabrication of the sensor can be effectively translated to a point-of-care diagnostic device, provided validation with real patient samples is demonstrated [45].

Recently, a SPCE with a nanointerface comprising bovine serum albumin (BSA), multi-walled carbon nanotubes (MWCNTs), and glutaraldehyde (GA) was reported for measurement of HbA1c. The BSA-MWCNT composite in PBS was crosslinked to the electrode by using glutaraldehyde, followed by immobilization of APBA using EDC-NHS coupling. The BSA imparted an anti-fouling nature to the electrode surface. The fabricated sensor detected HbA1c between 2 and 15% in the samples. To improve the sensitivity of the sensor, the working electrode was further modified by incorporation of the anti-HbA1c antibody. The sensor also evaluated unprocessed human serum [46]. However, the use of antibodies on the electrode surface could lead to increased costs and a reduced shelf-life.

**Table 1 sensors-23-01901-t001:** Boronic acid-based electrochemical sensors for direct detection of HbA1c.

Electrochemical Technique	Interface	Detection Range (HbA1c)	LoD	Ref.
**Amperometric**	APBA-pTTBA/AuNP/SPCE	0.1–1.5%	0.052%	[35]
	APBA/RVC	0.2–12 mg/mL	89 µg/mL	[34]
	MPBA/AuNF/SPCE	2–20% 5–1000 μg/mL	0.65%	[36]
	FPBA/PAMAM/DTSP/Au (GOx Backfilling)	2.5–15%	NA	[37]
	TiO_2_(G) NW@PAPBA–Au/ITO	2–10%	0.17%	[47]
	AQBA/SPCE	31.2–500 nM	4.2 nM	[40]
	ZrO_2_-DDAB/PG	6.8–14%	NA	[33]
	PAPBA/SPCE	0.975–156 μM	NA	[39]
**Impedometric**	APBA/ESM	2.3–14%	0.19%	[41]
	APBA/ESM	2.3–14%	0.21%	[48]
	APBA/GA/Cys/Au	0.1–8.36%	0.024%	[42]
	T3BA/Au	4.5–11.5%	NA	[43]
	GO-APBA/GCE	8%	NA	[44]
	T3BA	10–100 ng/µL	1 ng/µL	[45]
	APBA/BSA/MWCNTs/GA/SPCE	4–15%	1.2%	[46]

NA: Not applicable.

### 2.2. Electrochemical Immunosensors for HbA1c Detection Using Antibodies

Antibodies against HbA1c have been employed as specific capture agents to selectively retain HbA1c on the electrode surface. The captured HbA1c could result in increased impedance or could be quantified using an enzyme-linked secondary antibody for generating the electrochemical response. Molazemhosseini et al. [49] developed a disposable electrochemical immunosensor for the quantification of HbA1c using a three-electrode system (Figure 4). A polyethylene terephthalate (PET) sheet served as the flexible substrate for the electrode system, over which the Ag/AgCl reference electrode was printed as a thick film, while gold films deposited by sputtering formed the working and counter electrodes. The gold electrode surface pretreated sequentially in KOH, H_2_SO_4_, and HNO_3_ solutions was modified with a SAM of 3-mercaptopropionic acid (MPA) that served to immobilize anti-HbA1c antibodies via EDC-NHS coupling. The detection of HbA1c using differential pulse voltammetry (DPV) resulted in a linear range of 7.5–20 μg/mL. The sensor was validated in undiluted serum samples, where the linear range changed to 100–250 μg/mL. However, the reported range for diabetic individuals is greater than 9 mg/mL. Therefore, it is evident that the fabricated sensor needs additional modifications to enable detection of HbA1c in the clinically relevant range.

In another interesting approach for the quantification of HbA1c using anti-HbA1c IgG antibodies, a mixed layer of oligo (phenyl ethynylene) molecular wires (MW) and an oligo (ethylene glycol) (OEG) were employed as the interface. The distal end of the MW was modified by non-covalent addition of the HbA1c analogue glycosylated pentapeptide (GPP) and the redox mediator 1,1′-di(aminomethyl)ferrocene (FDMA). The sensor used the competitive inhibition assay for the quantification of HbA1c, where the surface-bound GPP and free HbA1c compete for the antibodies in the solution. The binding of the antibodies to the interface resulted in an attenuation in the ferrocene electrochemical response. Higher concentrations of HbA1c resulted in an increase in Faradaic currents due to lesser anti-HbA1c on the electrode interface, and vice versa. The measurement of the change in charge transfer resistance (R_ct_) was used for the quantification of the HbA1c between 4.5 and 15.1% [50]. The same group also reported another method to quantify HbA1c based on EIS measurements [51]. In this method, the glassy carbon electrode was modified with 4-aminophenyl (Ph-NH_2_) and AuNP. The unreacted amine groups of (Ph-NH_2_) were blocked with OEG, followed by surface modification of AuNP with Ph-NH_2_, and attachment of GPP (Figure 5). The fabricated sensor upon exposure to anti-HbA1c IgG antibodies resulted in a corresponding change of R_ct_ of the system due to the selective interaction of anti-HbA1c antibodies and GPP. The sensor had a wide detection range between 0 and 23.3% HbA1c and the values were in agreement with the conventional clinical techniques when tested with human samples. In another variant, a label-free amperometric immunosensor, using a modified interface with the redox mediator FDMA between the GPP- and the Ph-NH_2_-modified AuNPs, was also reported for detection of HbA1c based on the competitive immunoassay technique, where the linear detection range was between 4.6 and 15.1% in human blood samples [52].

Ion-sensitive field-effect transistor (ISFET)-integrated chip-based immunosensors have been investigated for quantification of HbA1c [53]. The detection setup used two gold working electrodes, two gold reference electrodes, and a platinum pseudo-reference electrode. The gold reference electrodes were modified by electropolymerization of polypyrrole (PPy), while the working electrodes were first modified by electropolymerization of PPy–HAuCl_4_ composite film, followed by in situ deposition of gold nanoparticles. Subsequently, the working electrodes were incubated in diluted antibody solution to immobilize the antibodies on the working electrode. The sensor detected both HbA1c and Hb and the working range of the sensor was between 5 and 20% HbA1c. When the interface was changed to SAM-modified gold nanospheres that were conjugated to mercaptamine-modified gold electrode, followed by immobilization of antibodies to SAMs using EDC-NHS chemistry, the sensor was able to detect HbA1c between the linear range of 50 and 170.5 ng/mL and total Hb between 166.7 and 570 ng/mL [54]. The same strategy was employed to develop a field-effect transistor chip with electrodes surface-modified with 1,6-hexanedithiol and a nano-gold monolayer linked with anti-HbA1c antibodies. The sensor detected Hb and HbA1c between the ranges of 60 and 180 μg/mL and 4 and 24 μg/mL, respectively [55]. The same group had reported a FET-based device where the gold nanofilm was grown by a seed-mediated technique over the gold electrode containing a mixed SAM. After conjugation of the antibodies to the gold nanoparticles, the FET sensor was employed for detection of Hb and HbA1c. The sensor showed a linear range of 1.67–170.5 ng/mL and 16.7–1705 ng/mL for HbA1c and Hb, respectively [56]. The presence of the nanogold layer was found to confer additional sensitivity towards the detection. These studies suggest that apart from the chemical nature, the mode of formation of the interface could also influence the detection range and sensitivity of the resultant sensor.

A modified sandwich immunoassay protocol with an anti-Hb molecule serving as the capture antibody specific to the total Hb, and an enzyme-labeled detection antibody specific for HbA1c, was studied for quantification of HbA1c. Though four different anti-Hb and two different anti-HbA1c antibodies were investigated in the study, none of the results were in agreement with the actual HbA1c values of the samples. It was hypothesized that the detection site of the anti-HbA1c antibody is hidden inside due to folding, and hence an additional denaturation step using 0.2% dodecyltrimethylammonium bromide was introduced in the protocol to expose the antigen-binding site of the anti-HbA1c antibody. This resulted in improved sensitivity of the immunosensor, that was now able to quantify HbA1c between 5.6 and 10.6% [57]. Further tuning of the sensitivity may be required for potential clinical deployment of the sensing device in the future.

Another study employed the sandwich immunoassay technique for dual-detection of HbA1c through electrochemical and optical methods, where the nano-bioprobe was obtained by conjugation of cadmium telluride (CdTe) quantum dots to the secondary antibody. The sandwich immunoassay was performed using an anti-HbA1c antibody as the capture antibody along with the quantum dot-labeled secondary antibody as a dual-mode sensor for the quantification of HbA1c. The sensor employed a small volume of the sample diluted 500 times for the detection and exhibited linearity between 4 and 16% HbA1c [58]. In this line, a bovine serum albumin (BSA)-MWCNT composite crosslinked to a screen-printed carbon electrode by glutaraldehyde was employed as a nano-biosensing interface for the quantification of HbA1c [46]. The anti-HbA1c antibodies were conjugated to the interface along with APBA using EDC-NHS coupling. The resultant immunosensor quantified HbA1c between 2 and 15% in blood samples. The electrode also exhibited good stability over a month when stored in BSA with less than an 8% loss in signal response. Recently, ratiometric studies related to detection of both HbA1c and Hb have been reported [59]. This methodology relied on the sandwich immunoassay, which was assisted by acridine ester-labeling, and the %CV for both repeatability for multiple detections and within-site reproducibility were documented as 1.22 to 2.21% and 2.13 to 3.27%, respectively.

## 3. Aptamer-Based Sensors for HbA1c Detection

Aptamers, also known as chemical antibodies, are short, single-stranded sequences of DNA, RNA, or peptides that display very high specificity towards a specific target, similar to those displayed by antibodies. However, they exhibit better stability and are less expensive that their antibody counterparts. These advantages have resulted in their use as capture agents in biosensors [60]. Several aptamer-based sensors have been reported in the literature for the electrochemical detection of HbA1c (Table 2). A paper-based nano-geno sensor for the determination of HbA1c in blood was fabricated using a composite interface of reduced graphene oxide (GO) and gold nanoparticles, followed by immobilization of the thiolated aptamer over the graphite electrode (GS/rGO-Au). In order to prevent non-specific binding, the free binding sites were blocked using 11-mercapto-1-undecanol (MU) solution. The sensor was further evaluated for its stability, selectivity, and real sample quantification using the differential pulse voltammetry (DPV) technique in the presence of a ferro/ferricyanide redox couple. The sensor displayed a broad linear range of 1 nM–13.83 μM of HbA1c and a high sensitivity of 269.2 μA/cm^2^ [61].

In another study, a thiolated aptamer-based sensing label-free detection platform was employed for the detection of HbA1c using an aptamer-conjugated gold electrode. The resulting modified sensor was employed for label-free detection of HbA1c and total Hb in blood using square wave voltammetry. The sensor exhibited a linear range between 100 pg/mL and 100 ng/mL for both Hb and HbA1c. The sensor was also validated using human blood samples, where it detected HbA1c between 6.6 and 10.5% [62]. The same group also conducted a study comparing six different carbon nanointerfaces, namely, carbon (C), graphene (G), graphene oxide (GO), multi-walled carbon nanotube (MWCNT), single-walled carbon nanotube (SWCNT), and carbon nanofibers (CNF), for the aptamer-based determination of HbA1c. The aptamers were non-covalently immobilized on the nano-interfaced electrodes via π–π stacking interactions between the DNA aptamer and the carbon nanomaterial, which restricts the electron transfer. However, upon binding of the target protein, the aptamer dissociates from the surface, resulting in enhanced electron transfer, which was quantified as a measure of HbA1c. The results revealed that the SWCNT-modified sensor showed a superior response when compared to other interfaces in the presence of a ferro/ferricyanide redox couple. The SWCNT displayed an ultra-low LoD of 0.03 pg/mL for HbA1c and a detection range of 0.1 pg/mL to 1000 ng/mL. The sensor also displayed excellent specificity and its electrochemical response remained unaffected by the presence of other proteins encountered in blood, such as STAT3, DOCK8, SMN, and CFTR [63].

In an interesting strategy, a non-enzymatic aptasensor has been fabricated using bacteriorhodopsin-containing purple membrane isolated from bacteria. These were modified with aptamers against Hb and HbA1c. The binding of the protein results in a decrease in the photocurrent measured on the electrode, which was used to quantify the amounts of Hb and HbA1c in the sample. The sensors displayed a dynamic linear range of 0.1–100 μg/mL for both Hb and HbA1c. Additionally, the Hb and HbA1c aptasensors displayed excellent stability of 14 and 28 days, respectively. The sensor performance was validated using 19 human blood samples and the results showed a good agreement with conventional measurements. The detection time of the sensor is 15 min [29]. Further fine-tuning of the fabrication parameters are required for reducing the response time to under a minute for applications as a point-of-care diagnostic device.

## 4. Non-Enzymatic HbA1c Sensors Based on FV Detection Principle

Several non-enzymatic amperometric sensors have also been developed for FV-based HbA1c detection. FcBA-based quantification of FV was attempted using a glassy carbon paste electrode (GCPE) as well as with a glassy carbon electrode (GCE) as the working electrode. It was reported that 3 mM of FcBA added to phosphate buffer solution successfully detected the varying concentrations of FV with a LoD below 0.05 mM at a low oxidation potential of +0.1 V [64]. In a novel strategy, the catalytic property of polyvinylimidazole (PVI) was employed as an artificial dehydrogenase sensor to detect FV. The PVI was immobilized in the carbon paste electrode and used for amperometric detection studies, where a linear range of 20 to 700 μM was achieved. Colorimetric detection of FV was also demonstrated by mixing PVI, meta-phenazinemethosulphate (m-PMS), dichlorophenol indophenol (DCIP), and Triton X-100 in phosphate buffer of pH 7.0. The reduction in absorption intensity of DCIP upon FV addition was quantified at 600 nm. The linear range of detection by this method was wider, between 50 μM and 10 mM [65]. The sensitivity of detection of the electrochemical approach was better but the range of detection was limited, which was attributed to the poor performance of the redox mediator employed in the study, namely m-PMS.

Molecular imprinting techniques that create a recognition site for a specific analyte on a substrate, that is usually a polymeric matrix, have been employed for development of molecularly imprinted polymeric (MIP) sensors. In electrochemical MIP sensors, specific capture and subsequent quantification of the captured analyte are achieved through monitoring the changes in the electrochemical properties. The MIP sensors are a cost-effective method for quantification with high specificity of the molecule of interest. MIP sensors have been explored for measurement of FV as a non-enzymatic option for HbA1c detection. A potentiometric sensor with the molecular imprints of FV in poly-aminophenyl boronic acid (PAPBA) was deposited over indium-doped tin oxide (ITO) electrodes. The molecularly imprinted PAPBA sensor demonstrated good selectivity for the fructose moieties of the Amadori compound [66]. Further studies using this sensor on human samples is warranted to understand its potential as a reliable HbA1c sensor. Recently, a novel dual-imprinted polymer-based flexible electrochemical sensor for diabetic markers was fabricated from flexible carbon paste aluminum foil modified by electro-polymerization of dual molecularly imprinted poly(rhodamine B) along with PAPBA nano-cubes. While poly(rhodamine B) was used for non-covalent binding of hemoglobin, PAPBA was employed for cis-diol interaction-based specific detection of HbA1c [67]. Despite the molecular imprints produced on PAPBA being relatively cheap, subsequent studies by other groups have revealed that these sensors are best for single-use devices, as their protein recognition ability deteriorates after the first use [68].

## 5. Biosensors Based on FV Detection

Antibody-based immunosensors for HbA1c are expensive, with a poor shelf-life and greater susceptibility to loss of function due to denaturation of the antibodies. The use of enzyme-based sensing elements offers the advantage of specificity and a robust electrochemical response that can be directly correlated with the analyte concentrations (Table 3). Fructosyl amino acid oxidase (FAO) has been isolated from both bacterial sources, such as *Corynebacterium* and *Agrobacterium*, as well as fungal species, such as *Aspergillus*, *Penicillium*, *Fusarium*, and *Pichia*. These differ in their substrate specificity, with the bacterial enzymes showing high specificity towards alpha glycated amino acids [69]. The fructosyl peptide oxidase (FPOX) enzyme isolated from fungal species such as *Eupenicillium terrenum* and *Coniochaeta* sp. has shown significant sequence similarity with FAO. It also catalyzes the oxidation of the fructosyl valine-histidine peptide (FV) obtained from the protease digestion of HbA1c to form glucose and valyl histidine, with the concomitant reduction of the FAD coenzyme to FADH_2_. The reoxidation of FADH_2_ occurs with the formation of H_2_O_2_ that is detected electrochemically, either directly or indirectly, in the presence of redox mediators (Figure 6).

Biosensors based on immobilization of the enzyme on suitable matrices offer benefits such as portability, reusability, and rapid detection of HbA1c, as well as being less expensive than immunosensors. Immobilization of enzymes on the electrode surface offers a solution to many issues, such as leaching of the enzyme, maintenance of enzyme stability, and the shelf-life of the sensor. The choice of immobilization strategy will therefore be a major influence on the sensing performance. Another facet that has emerged in recent decades towards improved performance of enzyme-based sensors is the fabrication of electrodes with nanointerfaces. This has resulted in additional advantages in terms of increased stability, mechanical strength, improved conductivity, promotion of catalytic transformation of redox species, and inhibition of oxidation of interferants, thereby improving the signal-to-noise ratios. Nanoparticles also aid in increasing the electro-active surface area, promoting increased loading of the enzyme, and leading to an enhanced electron transfer at the electrode–electrolyte interface. Various electrochemical sensors with different electron conducting interfaces and enzyme immobilization matrices have been designed for HbA1c detection using the FV method. The sensing performance of these enzyme-based HbA1c biosensors is dependent upon the efficiency of enzyme immobilization as well as the source of the enzyme employed, which in turn influences the specificity of detection. Most of these amperometric biosensors exhibit a response time within 120 s between the pH window of 7.0 and 7.5, which is in the physiological pH range. Most electrochemical amperometric sensors reported in the literature could be operated in the temperature range of 25 °C to 45 °C, with a linear range of detection between 0.1 mM and 700 mM [70].

In an extensive study, a FAO enzyme extracted from marine yeast *Pichia* sp. N1-1 strain was immobilized in a special water-soluble polymer, namely, poly(vinyl alcohol)–stylbazole (PVA–SbQ), and the resultant thin film was deposited over a platinum working electrode and photo-crosslinked. When used for quantification of HbA1c in the presence of the redox mediator methoxy phenazine methosulfate, this sensor exhibited a detection range of 0.05–1.8 mM and a sensitivity of 0.42 μAmM^−1^. When FAO was immobilized in a carbon paste electrode and crosslinked using 1% glutaraldehyde, the sensing range obtained for HbA1c in the presence of the redox mediator was between 0.2 and 2.7 mM, with a sensitivity of 1.1 μAmM^−1^ [71]. Other redox mediators, such as ferrocene with peroxidase and Prussian blue, were also investigated for quantification of HbA1c on the PVA-SbQ/Pt electrode. The linear range obtained in both cases was only between 0.1 and 0.3 mM [72]. These results clearly indicate that the nature of the redox mediator also plays a critical role in the detection characteristics of the working electrode.

Use of nanointerfaces has resulted in an improved response and lesser loss of the electrons generated during the redox reactions at the electrode–electrolyte interface. Some of these nanoparticles with high surface area to volume ratios also provide immobilization sites for the enzymes, resulting in a higher active surface area. Several types of nanointerfaces have been explored both independently and as composites in electrochemical sensors for quantification of HbA1c. In one such study, the FAO enzyme was immobilized in an interface comprising zinc nanoparticles dispersed in polypyrrole film over a gold electrode (FAO/ZnONPs/PPy/Au) and employed for the quantification of HbA1c [73]. The sensor exhibited a LoD of 50 μM and a wide linear range between 0.1 and 3 mM. Other interfaces containing core-shell magnetic bio-nanoparticles (MNP) coated with the amine-containing polymer chitosan have also been studied as immobilization matrices for the FAO enzyme over gold electrodes (FAO/MNPs/Au) for the quantification of HbA1c [74]. The sensor exhibited a linear range up to 2 mM, with a rapid response time below 4 s and a LoD of 100 μM. The biosensor exhibited good reusability up to 250 measurements over a period spanning 3 months when maintained at a temperature of 4 °C to prevent enzyme denaturation. However, the clinically defined high levels of HbA1c are above 40 mM and hence the linear ranges of these sensors need further improvement for clinical relevance. Multiple nanostructures comprising gold nanoparticles (AuNPs) entrapped in phosphotungstic acid-coordinated TiO_2_ nanotubes have been investigated as a composite interface to facilitate electron transfer at the electrode–electrolyte interface (FAO/GNPs-PTA-TiO_2_/ITO) for an improved sensing response for the quantification of FV (Figure 7) [75]. The sensor exhibited a linear range of 0.5 μM to 2 mM with a response time of 3 s and a LoD of 0.5 μM. The sensor performance was found to be unaffected in the presence of interferents such as bilirubin, urea, cysteine, triglycerides, and ascorbic acid, which resulted in less than a 10% response from the working electrode. However, both uric acid and glucose elicited an electrochemical response between 15 and 18% from the working electrode. Though a rapid response was achieved, the linear range requires further improvement and a lower response to glucose for clinical use.

Along similar lines, FAO bio-conjugated to gold and platinum nanoparticles was electrodeposited over a poly(indole-5-carboxylic acid)-modified gold electrode. The authors evaluated the fabricated biosensor for its stability, and it was found that the sensor retains 50% of its initial activity until 12 weeks after fabrication [76]. The sensor did not respond to interferents such as bilirubin, urea, cysteine, glucose, ascorbic acid, and triglycerides, and displayed a linear range between 0.1 μM and 1 mM of HbA1c. The biosensor was also successfully tested in human blood samples for HbA1c measurement using a ferro/ferricyanide redox couple. The sensor, however, showed a >10% response to uric acid. The same group had also reported a nanointerface with Au nanoparticles (AuNPs) embedded in N-doped graphene nanosheets (GN) with the FAO enzyme coated on a fluorine-doped tin oxide (FTO) glass electrode (FAO/AuNPs/GNs/FTO). This modification showed promising results in the electrochemical detection of FV as an indirect measure of HbA1c between the concentration range of 0.3 μM and 2 mM [77]. In another variant, reduced graphene oxide (rGO) mixed with multiwalled carbon nanotubes (MWCNT) was electrodeposited over a gold (Au) working electrode, followed by electrodeposition of platinum nanoparticles and drop-casting of FAO to form the working electrode. The FAO/PtNPs/rGO–MWCNT/Au was successfully employed for quantification of HbA1c between 0.05 and 1000 μM, with a LoD of 0.1 μM [78].

Screen-printed electrodes have emerged as a portable, flexible, and disposable option for clinical diagnosis. In a proof-of-concept study, recombinant FAO from *E. coli* was immobilized over an iridium-carbon ink screen-printed electrode that was employed for measurement of FV with a sensitivity of 0.0215 mAmM^−1^ cm^−2^ [79]. Along similar lines, a single-use disposable screen-printed electrode setup for FV and HbA1c was developed by drop-casting FAO on graphite ink screen-printed electrodes [80]. The fabricated disposable sensor was employed for the quantification of FV and was also evaluated for its sensing performance using human blood samples. The biosensor showed effectiveness for quantification of fructosyl valine (FV) as well as human blood samples, with a wide linear range of 1 to 8 mM and a sensing time of 60 s.

Recently, fructosyl peptide oxidase (FPO or FPOX)-based HbA1c sensors have also gained traction as a substitute for the more extensively studied FAO enzyme. A composite interface comprising chitosan (CHIT), graphene oxide (GO), and gold nanoparticles (AuNPs) was employed to immobilize FPOX by drop-casting to obtain the final working electrode, FPOX/CHIT-GO-AuNPs/FTO [81] (Figure 8). The sensor displayed a LoD of 0.3 μM and quantified FV between 0.1 and 2 mM. The performance of the sensor was studied using human blood samples and the values were found to be in agreement with those obtained using a commercially available FPOX-based biochemical assay kit.

A dual-enzyme-based microfluidic chip was fabricated for the detection of glucose and HbA1c. Gold electrodes patterned by photolithography were treated with cysteamine followed by enzyme conjugation through glutaraldehyde crosslinking. Glucose oxidase and FAO were employed for the specific detection of glucose and HbA1c, respectively. The chip employed di-electrophoresis for separation of blood plasma and had separate detection areas for measurement of glucose and HbA1c that regulated sample movement by micropumps and microvalves. The sensor detected glucose between 40 and 200 mg/dL and HbA1c in the range 0–7% [82]. This automated chip with slight modifications to extend the detection range of HbA1c may find use for clinical diagnosis in the future.

**Table 3 sensors-23-01901-t003:** Amperometric biosensors for detecting HbA1c based on FV.

Transducers	Detection Range (FV)	Sensitivity	LoD	Ref.
FAO/Carbon paste (in methoxy phenazine methosulfate)	0.2–2.7 mM	1.1 µA mM^−1^ cm^−2^	NA	[71]
FAO-(PVA-SbQ)/Pt	0.05–1.8 mM	0.42 µA mM^−1^ cm^−2^	NA	[71]
FAO-POD-Ferrocene-(PVA-SbQ)/Pt	0.1–0.3 mM	0.4 µA mM^−1^ cm^−2^	NA	[72]
FAO-(PVA-SbQ)/PB /GCE	0.1–0.3 mM	1.3 µA mM^−1^ cm^−2^	NA	[72]
FAO-(PVA-SbQ)/Pt (flow injection analysis)	0.2–10 mM	4.6 nA mM^−1^ cm^–2^	NA	[83]
FAO/ZnONPs/PPy/Au	0.1–3.0 mM	38.42 μA mM^−1^	50 µM	[73]
FAO/MNPs/Au	0–2 mM	NA	0.1 mM	[74]
FAO/GNPs-PTA-TiO_2_/ITO	0.5–2000 μM	NA	0.5 µM	[75]
FAO/AuNPs-PtNPs/PIN5COOH	0.1–1000 μM	0.316 μM/mA	0.2 µM	[76]
FAO/AuNPs/GNs/FTO	0.3–2000 μM	NA	0.2 µM	[77]
FAO/PtNPs/rGO–MWCNT/Au	0.05–1000 μM	0.562 mA/μM	0.1 µM	[78]
FAO/Iridium-Carbon ink	NA	21.5 μA mM^−1^ cm^−2^	NA	[79]
FAO/Graphite ink	0–8 mM	NA	NA	[80]
FPOX/CHIT-GO-AuNPs/FTO	0.1–2 mM	8.45 µA mM^−1^ cm^−2^	0.3 µM	[81]
GCPE/ITO	0–1 mM	5.26 μA mM^−1^	0.05 mM	[64]
Carbon–PVI (in methoxy phenazine methosulfate)	20–700 μM	NA	20 μM	[65]

NA: Not applicable.

An electro-chemiluminescent biosensor fabricated using a composite nanointerface comprising Au nanoparticles and titania nanotubes (TiNT) with immobilized FAO coated over an ITO electrode was employed for detection of FV. The H_2_O_2_ released by the enzyme-catalyzed redox reaction was used to convert luminol, resulting in emission of electromagnetic radiation that was further used to enhance the electrochemical response. The sensor exhibited a linear range of detection between 4.0 nM and 0.72 μM of FV and an ultrahigh sensitivity with a detection limit of 3.8 nM. The sensor retained its sensing performance over 20 days when stored at 4 °C. The sensor showed good correlation with the results obtained using a commercial HbA1c enzymatic kit when tested with human blood samples [84]. Recently, a label-free affinity sensor based on a field-effect transistor (FET) was fabricated for the detection of FV. The interface consisted of single-walled carbon nanotubes (SWCNT) that served as the immobilization surface for the bio-recognition element fructosyl amino acid-binding protein (SoCA), isolated from *Rhizobium radiobacter*. The SoCA is a bacterial protein about 29 kDa in molecular weight and has a high specificity towards α-fructosyl amino acid. The SWCNT also served as a semi-conducting transducing element. The sensor exhibited excellent specificity towards FV with a low detection limit of 1.2 nM [85]. Recently, tungsten disulfide nanoparticle (WS_2_ NP)-enhanced MIP-based material was utilized for the HbA1c sensing in diabetic patients. The resultant biosensor showed a detection limit at 0.01 pM and the sensitivity was noted to be 0.27 µA/pM [86] (Figure 9).

## 6. Multi-Analyte Sensors

In an effort to overcome the limitations accompanying the diagnosis of diabetes based solely on HbA1c measurements, multi-analyte measurements have been gaining traction recently. One strategy is to simultaneously measure total Hb and HbA1c. Not only does this strategy reduce errors owing to differences in basal Hb levels, it can also provide information on the anemic status of the individual. There are certain sensors designed to detect both the total Hb (tHb) and HbA1c in the blood sample, keeping in mind the fact that in real samples, the HbA1c is measured in terms of HbA1c% to total Hb. Several potentiometric field-emission transmitter (FET) sensor chips fabricated by Micro-Electro-Mechanical-Systems (MEMS) and Complementary Metal-Oxide-Semiconductor Transistor (CMOS) techniques have been reported to quantify both tHb and HbA1c [53,54,55,56]. In a recent study [62], an aptasensor employed a novel interface with aptamers displaying high affinity for tHb and HbA1c for quantification.

A three-dimensional paper-based electrochemical impedance device (3D-PEID) has been reported for simultaneous sensing of tHb and HbA1c [48]. The electrode surface comprised egg-shell membranes (ESM) modified with APBA and haptoglobin to detect HbA1c and tHb, respectively. Liu et al. [34] designed an interface by using chitosan (CS) and tetraethyl silica (TEOS) to immobilize the carbon nanotubes on screen-printed electrodes for tHb detection, while HbA1c was detected using an APBA-modified reticulated vitreous carbon (RVC) electrode.

In another study, molecular imprinted conducting nano-cubes were employed to detect HbA1c and tHb. The sensor platform contains dual molecularly imprinted nano-cubes with PAPBA for HbA1c, along with poly-rhodamine B (for tHb) deposited on carbon paste-coated aluminum foil for detection [67]. Recently, Thiruppathi et al. [40] developed a simple platform for detecting tHb (in blood samples) by immobilizing MWCNT in SPCE using Nafion, while HbA1c in diluted blood samples can be detected in a separate modified platform.

Measurement of HbA1c along with other diabetic markers may offer higher diagnostic accuracy, for instance, a novel heterojunction nanohybrid interface comprising electrospun TiO_2_ nanowires and carboxylic acid-functionalized graphene (G-COOH), along with a composite of gold nanoparticles (Au NPs) and PAPBA. The nanointerface was denoted as TiO_2_(G)NW@PAPBA-Au HJNH and was drop-cast on a pre-cleaned indium tin oxide (ITO) electrode, and the working electrode was used to non-enzymatically detect glucose by the photoelectrochemical method and HbA1c through the electrochemical method [47]. The analytes from the sample were captured on the electrode surface by PABPA. The detection range for glucose was between a wide range of 0.5 and 28 mM, while the sensor was able to quantify HbA1c between 2 and 10% with a LoD of 0.11 mM and 0.17%, respectively. Though two key markers of diabetes were detected by this sensor, the upper limit of detection for HbA1c could be extended for better clinical utility. Further, the sensor requires clinical validation for establishment of its diagnostic utility in clinics. Recently, an electrochemical PoC device with disposable electrodes modified with chemical capture agents was successfully employed to quantify Hb, HbA1c, and glycated albumin from serum samples, as well as microalbumin and creatinine from urine samples. This integrated detection device enabled superior diagnostic efficiency for anemia, diabetes, and diabetic kidney disease [87].

## 7. Strategies for Measurement of HbA1c

Ever since Anthony Cerami introduced the first measurement method for HbA1c in the 1980s, it has been widely implemented in the clinics for diabetes diagnosis and management. Over the years, additional methods have been introduced for HbA1c quantification. These include chromatography-based methods such as ion exchange chromatography [88], high-performance liquid chromatography (HPLC) [89], and boronate affinity chromatography [90]. Apart from these, other methods such as immunoassay, enzyme-linked immunosorbent assay (ELISA), colorimetric method, liquid chromatography combined with mass spectroscopy, capillary electrophoresis, and fluorometry have been used for HbA1c detection. These techniques require expensive reagents and/or components that limit their utility for screening large population sets. Further, these instruments are not portable, and hence HbA1c measurements are mostly laboratory-based. Advancements in manufacturing technology have resulted in the development of some point-of-care (PoC) analyzers that work on these conventional techniques based on differences in charge (ion-exchange chromatography) and structure (boronate affinity chromatography and immunoassays). These include the commercially available DiaSTAT, DS5, and G7 analyzers that are based on cation-exchange chromatography, and the Vision Analyzer, NycoCard^®^, Clover, etc., which are based on boronate affinity chromatography (Table 4).

The main advantages of PoC analyzers include ease of use and instantaneous results. However, it must be ensured that these PoC analyzers estimate the quantities of HbA1c in the sample with high accuracy and specificity, and should be cheap and reproducible. A survey analyzing the proficiency of various clinical laboratory techniques employed for HbA1c measurement revealed that nearly 65% depend on an immuno-turbidometric assay for HbA1c measurement, 32% are based on ion-exchange chromatography, while >2% are based on boronate affinity chromatography [102]. Another study compared the efficacy of eight HbA1c PoC analyzers, namely, Quo-Test (Quotient Diagnostics, Surrey, UK), In2it (Bio-Rad, Hercules, CA, USA), NycoCard, (Abbott), Afinion (Abbott), A1CNow (Bayer, Berkeley, CA, USA), DCA Vantage (Siemens), Clover (Infopia, Köln, Germany), and InnovaStar (DiaSys, Holzheim, Germany). The results suggested that DCA Vantage and Afinion complied with the criteria in the National Glycohemoglobin Standardization Program (NGSP), with a total imprecision (CV) below 3% [103]. Another comparative study evaluated the performances of the PoC analyzers Afinion and DCA Vantage with the HPLC technique. The study concluded that both the PoC and HPLC methods exhibited certain biases and overestimated HbA1c in a certain window when compared to NGSP standards [104]. Interestingly, in a comparative study of the PoC analyzer Cobas b 101 (Roche Diagnostics) and HPLC [105] for HbA1c quantification from EDTA whole blood and capillary blood, both techniques yielded similar results, thus giving the PoC system an edge owing to its ease of usage. Recently, in 2018, another study was conducted to evaluate the performance of four PoC analyzers, which was compared with four secondary reference measurement procedures (SRMPs), recognized by the International Federation of Clinical Chemistry and Laboratory Medicine (IFCC) and the National Glycohemoglobin Standardization Programme (NGSP). The four PoC were Afinion2™ (Abbott, boronate affinity separation), Quo-Lab (EKF Diagnostics PLC, boronate affinity separation), HbA1c 501 (HemoCue Diagnostics, boronate affinity separation), and A1Care (i-SENSE, enzymatic determination). The results revealed that the Afinion2 and Quo-Lab devices performed the best. It is evident from these studies that most of these PoCs are limited by their sensitivity, response time, need for expensive reagents, and lack of specificity [106]. Hence, there is an acute need for new detection strategies that can overcome the disadvantages of the conventional methods and offer rapid, sensitive, and cost-effective measurement of HbA1c.

## 8. Challenges in Diagnosis Based on HbA1c Measurements

The total Hb values vary between individuals, and hence measurement of HbA1c alone results in errors in the diagnosis of diabetes, especially in anemic individuals [107]. Therefore, it becomes necessary to measure both total Hb and HbA1c levels in individuals for error-free diagnosis. Another key challenge in diabetes diagnosis using HbA1c arises due to glycated hemoglobin variants that exist due to various congenital disorders of globin chain synthesis, called “hemoglobinopathies”. Some of these variants directly interfere with the HbA1c detection, leading to erroneous values, while some other variants cause interference by premature turnover of red blood cells (RBC). In cases where an individual is heterozygous, the individual will be asymptomatic and have normal red cell survival [108]. However, the presence of these variants can lead to the under- or over-estimation of glycated hemoglobin. Therefore, initial testing needs to be performed to screen the person for hemoglobinopathies before HbA1c measurements to avoid errors. Similarly, conditions leading to blood loss or hemolytic anemia have also been reported to result in erroneous HbA1c values [109]. Another facet in diagnosis using HbA1c levels is the variations in the predictions based on HbA1c with those based on blood glucose and glycated albumin levels, which are referred to as the hemoglobin glycation index (HGI) and glycation gap (GG), respectively [110]. Both glucose and glycated albumin are extracellular entities, while HbA1c formation is intracellular. Therefore, HGI and GG have been attributed to the difference in permeation of glucose through the red blood cell (RBC) membrane to glycate Hb, the rate and position of glycation and deglycation in the globin chains, as well as the lifespan of the RBCs where the non-enzymatic glycation takes place [111].

## 9. Concluding Remarks

Sensors are used in all domains of life, and hence there has been a continuous evolution of the sensing devices from being lab-based to portable point-of-care devices. In an effort to improve the sensitivity, selectivity, speed, stability, accuracy, and range of detection, newer materials and their combinations have been explored as interface materials and sensing or transduction elements. With the exponential increase in the diabetic population, there is an urgent need to develop point-of-care devices for quantification of diabetes markers towards efficient diagnosis and management of diabetes. The growing importance of glycated hemoglobin (HbA1c) as a diabetes biomarker has now opened up new avenues for the development of effective electrochemical sensor platforms for affordable commercial devices and translating them into point-of-care analyzers. In this review, we have summarized the recent developments in the past two decades towards fabrication of efficient sensors for the quantitative detection of HbA1c. A diverse combination of nanostructured metallic particles, metal oxides, inorganic complexes, carbon nanomaterials, quantum dots, and highly branched polymers, along with traditional redox partners, have been explored as interface layers and immobilization surfaces for the capture agent or the redox label towards the quantification of HbA1c or FV, a model compound for HbA1c. Direct detection of HbA1c is also being attempted through development of boronate-based sensors, aptamer-based sensors, or immunosensors. Emerging technologies such as 3D printing and microfluidics are being actively explored to transform clinical diagnosis from lab-based to portable PoC systems through systematic experimentation with newer combinations of nanointerface materials and deciphering the reactions occurring at the electrode–electrolyte interface. Current trends in the realm of sensors indicate a strong focus on translating the sensing technology into disposable paper electrode systems and PoC analyzers that can serve as ‘anywhere-anytime-anyone’ devices.

## Figures and Tables

**Figure 1 sensors-23-01901-f001:**
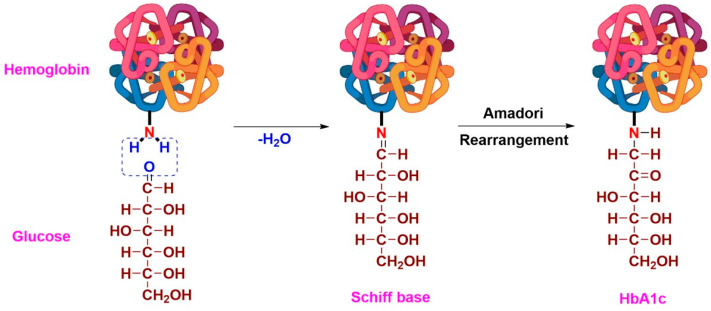
Scheme illustrating the formation of HbA1c by non-enzymatic glycation of hemoglobin via Amadori rearrangement.

**Figure 2 sensors-23-01901-f002:**
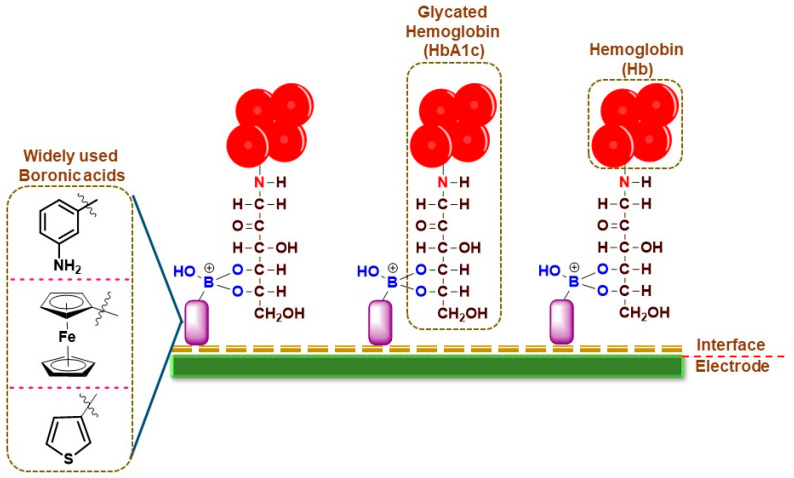
Pictorial representation of broadly used boronic acid-assisted electrochemical sensors for HbA1c detection.

**Figure 3 sensors-23-01901-f003:**
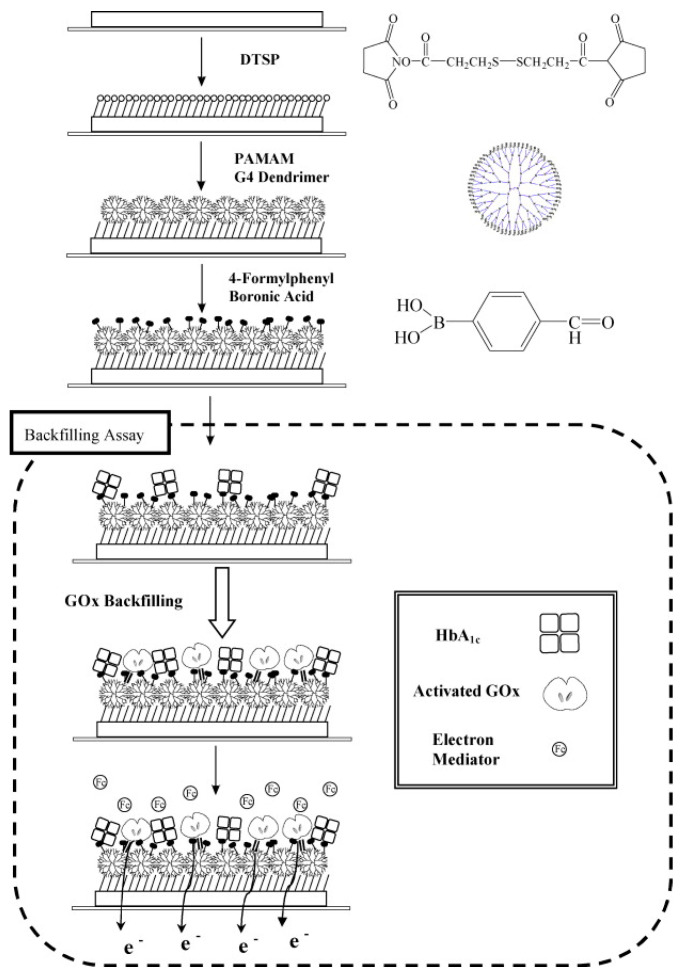
Scheme representing the backfilling assay−mediated amperometric detection of HbA1c employing boronic acid derivatives. Reprinted with permission from [37].

**Figure 4 sensors-23-01901-f004:**
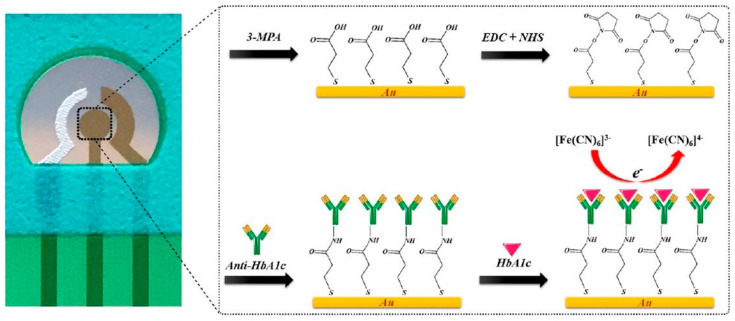
Electrochemical immunosensor based on gold as an interface for direct detection of HbA1c [49].

**Figure 5 sensors-23-01901-f005:**
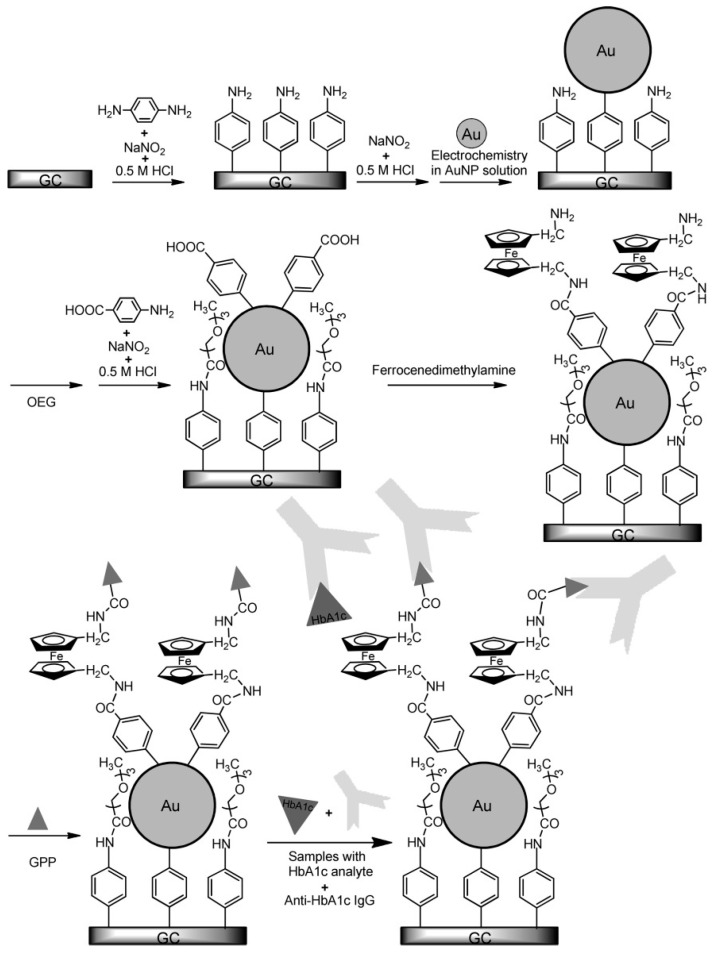
Immunosensor established by Liu et al. summarizes AuNP-assisted amperometric detection of HbA1c. Reprinted with permission from [52].

**Figure 6 sensors-23-01901-f006:**
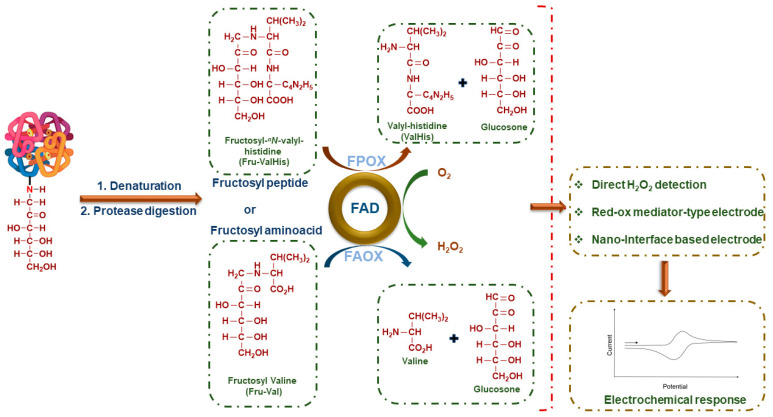
General representation of fructosyl amino acid oxidase (FAO) enzyme-based electrochemical sensors for HbA1c.

**Figure 7 sensors-23-01901-f007:**
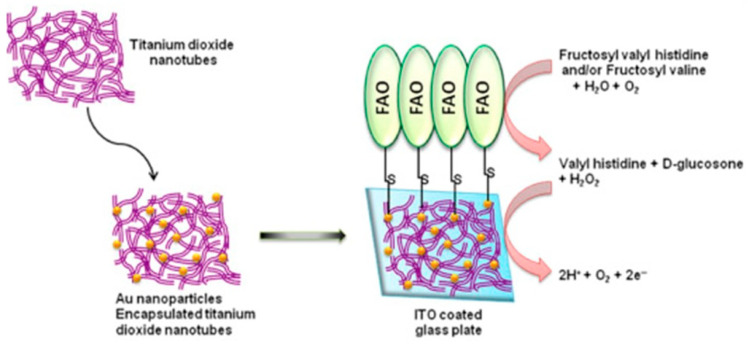
The final working electrode, FAO/AuNPs-PTA-TiO_2_/ITO, employed for quantification of FV. Reprinted with permission from [75].

**Figure 8 sensors-23-01901-f008:**
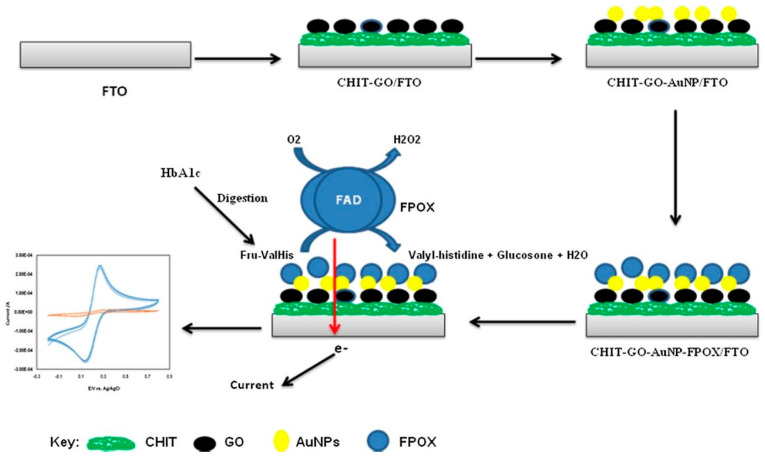
Modified working electrode FPOX/CHIT-GO-AuNPs/FTO for amperometric detection of FV. Reprinted with permission from [81].

**Figure 9 sensors-23-01901-f009:**
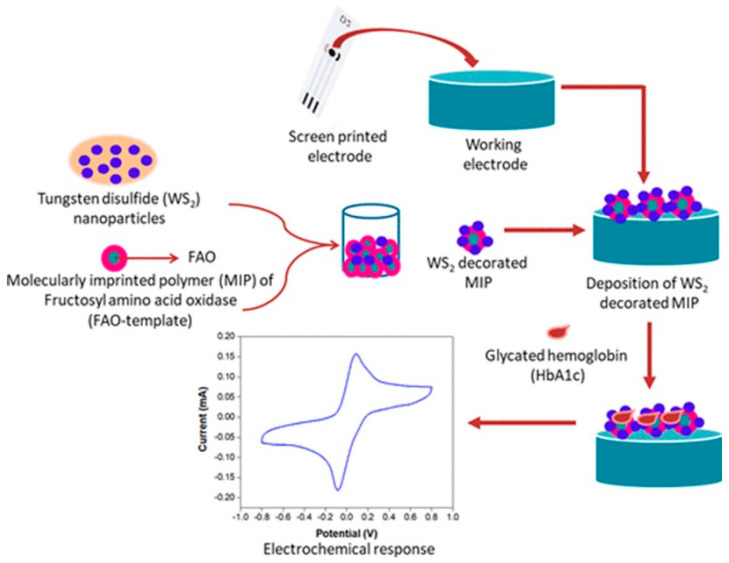
Schematic illustration of a WS_2_ NP−fabricated nano-biosensor based on the FAO principle for the HbA1c sensor in diabetic patients. Copyright CC BY−NC−ND 4.0 [86].

**Table 2 sensors-23-01901-t002:** Non-boronate biosensors for direct HbA1c detection.

Biosensors	Transducers	Type	Detection Range (HbA1c)	Sensitivity	LoD	Ref.
**Aptamer**	Aptamer/rGO-Au/GS	DPV	1–13.83 μM	269.2 μA·cm^−2^	1 nM	[61]
	Aptamer/AuNP/SPCE	SWV	100 pg/mL–100 ng/mL	NA	0.2 ng/mL	[62]
	Aptamer/SWCNT/SPCE	SWV	0.1 pg/mL–1000 ng/mL	NA	0.03 pg/mL	[63]
**Immunosensors**	Anti-HbA1c/MPA/Au	DPV	7.5–20 μg/mL (0.1 M PBS) 0.10–0.25 mg/mL (undiluted human serum)	NA	NA	[49]
	GPP/FDMA/MW/GCE (Competitive inhibition assay)	SWV	4.5–15.1%	NA	NA	[50]
	GPP/Ph-NH_2_/Au/Ph-NH_2_/GCE (Competitive inhibition assay)	Impedimetric	0–23.3%	NA	NA	[51]
	GPP/FDMA/4ABA/Au/Ph-NH_2_/GCE (Competitive inhibition assay)	Amperometric	4.6–15.1%	NA	NA	[52]
	Anti-HbA1c/Au/PPy/Au	Potentiometric	4–18 μg/mL	1.5087 Mv μg^−1^ mL	NA	[53]
	Anti-HbA1c/Mixed SAM-wrapped Au spheres/Au	Potentiometric	50–170.5 ng/mL	94.73 μV ng^−1^ mL	NA	[54]
	Anti-HbA1c/AuNP/SAM/Au	Potentiometric	4–24 μg/mL	−90.6 mV/[log(μg/mL)]	NA	[55]
	Anti-HbA1c/Mixed SAM/AuNP/Au	Potentiometric	1.67–72.14 ng/mL	165.2 μV ng^−1^ mL	NA	[56]
	CdTe	Anodic stripping voltammetry (ASV)	4–16%	NA	NA	[58]
	Anti-HbA1c/BSA/MWCNTs/GA/SPCE	Voltametric	2–15%	NA	0.4%	[46]

NA: Not applicable.

**Table 4 sensors-23-01901-t004:** List of various commercial PoC analyzers for evaluating HbA1c.

Manufacturer	Device Name	Method	Sample Volume (µL)	Detection Range	Analysis Time (s)	SRA Approvals	Ref.
Abbott (Scarborough, ME, USA)	Afinion 2	Boronate affinity	1.5	4.0–15.0%/ 20–140 mmol/mol	180	CE-IVD, FDA (CLIA waived), Japan	[91]
	NycoCard Reader II		5.0	4.0–15.0%/ 20–140 mmol/mol	180	CE-IVD	[92]
Aidian (Espoo, Finland)	QuikRead go	Immunoassay	1.0	4.0–15.0%/ 20–151 mmol/mol	350	CE-IVD	[93]
Arkray Inc. (Kyoto, Japan)	The Lab 001	Capillary electrophoresis	1.5	4.0–16.0%/ 20–151 mmol/mol	90	Japan	[93]
Boditech Med (Gang-won-do, Republic of Korea)	Ichroma II	Immunoassay	5	4.0–15.0%/ 20–140 mmol/mol	720	CE-IVD	[94]
DiaSys (Holzheim, Germany)	InnovaStar	Enzymatic	10	3.0–14.0%/ 9–130 mmol/mol	<300	CE-IVD	[93]
EKF Diagnostics (Penarth, UK)	Quo-Lab	Boronate affinity	4	4.0–15.0%/ 20–140 mmol/mol	240	CE-IVD	[95]
	Quo-Test					CE-IVD, FDA	[95]
Green Cross MEDIS Corp. (Yongin, Republic of Korea)	CERA STAT 2000	Boronate affinity	5	3.0–15.0%/ 9–140 mmol/mol	<180	CE-IVD	[96]
HemoCue (Ängelholm, Sweden)	HbA1c 501	Boronate affinity	4	4–14.0% 20–130 mmol/mol	300	CE-IVD	[91]
Human Gesellschaft für Biochemica und Diagnostica GmbH (Wiesbaden, Germany)	HumaMeter A1c	Boronate affinity	4	4.0–15.0%/ 20–140 mmol/mol	240	CE-IVD	[97]
i-SENS Inc. (Seoul, Republic of Korea)	A1Care	Enzymatic	2.5	4.0–15.0%/ 20–140 mmol/mol	260	CE-IVD	[93]
iXensor (Taipei City, Taiwan)	PixoTest	Immunoassay	5	4.0–15.0%/ 20–140 mmol/mol	180	CE-IVD, FDA	[98]
Jana Care (Bangalore, Karnataka, India)	Aina	Boronate affinity	5	4–15%/20–140 mmol/mol	180	CE-IVD	[99]
OSANG Healthcare (Gyeonggi-do, Republic of Korea)	Clover A1c Plus	Boronate affinity	4	4.0–14.0%/ 20–130 mmol/mol	300	CE-IVD, FDA	[93]
PTS Diagnostics (Whitestown, IN, USA)	A1cNow+	Immunoassay	5	4.0–13.0%/ 20–120 mmol/mol	300	CE-IVD, FDA (CLIA waived)	[100]
Roche Diagnostics (Risch-Rotkreuz, Switzerland)	Cobas b 101	Immunoassay	2	4.0–14.0%/ 20–130 mmol/mol	320	CE-IVD, FDA	[93]
Siemens Healthineers (Erlangen, Germany)	DCA Vantage	Immunoassay	1	2.5–14.0%/ 4–130 mmol/mol	360	CE-IVD, FDA (CLIA waived)	[93]
Wuxi BioHermes Biomedical Technology Co. (Wuxi, China)	A1C EZ 2.0	Boronate affinity	3	4.0–14.0%/ 20.2–129.5 mmol/mol	360	CE-IVD	[101]

## Data Availability

Not applicable.

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
