# Peer review of "Trends in Quantification of HbA1c Using Electrochemical and Point-of-Care Analyzers"

_sensors, 2023, doi:10.3390/s23041901_

Round 1
Reviewer 1 Report
The paper reviewed the current available technologies for the quantification of HbA1c using Electrochemical and PoC analyzers in details. The topic is very interesting, the authors did quite a lot of research and summary and provide a very comprehensive review. However, I believe that if manuscript can be better constructed in terms of the content arrangement, it could be bit easier for the readership to understand. Overall, I think the manuscript is ready for publishing at Sensors after minor revision. The detailed comments are below.
1. Content arrangement:
· “Strategies for measurement of HbA1c” reviewed the existing commercial products for HbA1c measurement. However I think it might be more appropriate to put it after all technical review, since all detection methods included in Table 1 are actually discussed in later sections. Also, for general readership, technical content might be more useful.
· “Challenges in diagnosis based on HbA1c measurements” – I think it might be better to have this right before the conclusion?
- Minor comments:
· Line 16, nanoparticulate should be nanoparticle?
· Line 22, two full stop mark.
· Line 653, typo “measurements”. Please do another spelling check.
· Please carefully examine all the abbreviation. (i.e, “tHb” shows ahead of the first full text, some abbreviation has been repeatedly explained in the text and after Table content. Please consider using appendix to concise.)
· Figure 5 readability low: text in figures is bit blurry. Please try improving figure resolutions.
· Please complete author contribution, and remove unused template instruction
· Please number the section to improve readability, it clarify which section is subsection, and what is the leading discussion for new topics.
· Please try expanding the figure caption so that the figure can be more self-explanatory
Author Response
Response to Reviewer-1 comments
1. Content arrangement:
Q.1 “Strategies for measurement of HbA1c” reviewed the existing commercial products for HbA1c measurement. However I think it might be more appropriate to put it after all technical review, since all detection methods included in Table 1 are actually discussed in later sections. Also, for general readership, technical content might be more useful.
Response: We thank the reviewer for the constructive suggestion. The section on “Strategies for measurement of HbA1c” has now been shifted in the revised manuscript as suggested.
Q.2 “Challenges in diagnosis based on HbA1c measurements” – I think it might be better to have this right before the conclusion?
Response: We agree with the suggestion of the reviewer. The content has been shifted in the revised manuscript as suggested.
2. Minor comments:
Q.3 Line 16, nanoparticulate should be nanoparticle?
Response: We thank the reviewer for pointing out the spelling error that has now been corrected in the revised manuscript.
Q.4 Line 22, two full stop mark.
Response: The comment is addressed in the revised manuscript.
Q.5 Line 653, typo “measurements”. Please do another spelling check.
Response: The typo error has been addressed (line 696 in revised manuscript).
Q.6 Please carefully examine all the abbreviation. (i.e, “tHb” shows ahead of the first full text, some abbreviation has been repeatedly explained in the text and after Table content. Please consider using appendix to concise.)
Response: We thank the reviewer for the constructive comment. The appendix mentioning abbreviations has been added to the revised manuscript.
Q.7 Figure 5 readability low: text in figures is bit blurry. Please try improving figure resolutions.
Response: The high resolution image has been provided for Figure 5 in the revised manuscript.
Q.8 Please complete author contribution, and remove unused template instruction
Response: The author’s contribution is now included and unused template sections have been removed as suggested.
Q.9 Please number the section to improve readability, it clarify which section is subsection, and what is the leading discussion for new topics.
Response: All sections have been numbered in the revised manuscript as suggested.
Q.10 Please try expanding the figure caption so that the figure can be more self-explanatory
Response: The captions for Figures (1-7) have been expanded in the revised manuscript as suggested.
Reviewer 2 Report
The author presented previous research works about HbA1c. in my opinion, the manuscript can be considered for publication after applying the comment below:
1. The email address of the corresponding author should have been added.
2. Most papers reported in this manuscript for HbA1C sensing are not new.
3. It is better for the author to report some parameters related to biosensors such as surface coverage of the bio-receptors (aptamer, antibody and etc), the K affinity of the biosensor and etc.
4. If a modifier like graphene, nanotube, gold nanoparticles and etc improved the analytical performances of the biosensors, the author should explain why that specific nanomaterial improves the analytical performances compared to other ones.
5. the number of the table is good enough for a review manuscript but the number of the scheme is not enough.
Author Response
Response to Reviewer-2 comments
Q.1 The email address of the corresponding author should have been added.
Response: The email address is provided now in the revised manuscript.
Q.2 Most papers reported in this manuscript for HbA1c sensing are not new.
Response: The revised manuscript has been updated with recent articles in this domain as suggested.
Q.3 It is better for the author to report some parameters related to biosensors such as surface coverage of the bio-receptors (aptamer, antibody and etc), the K affinity of the biosensor and etc.
Response: In the present context, most of the articles on electrochemical sensors for HbA1c have reported parameters like detection range, and sensitivity of the fabricated sensors while parameters like surface coverage and K affinity have been scarcely reported. Therefore, for maintaining uniformity of the article, we have not mentioned these parameters which were missing in most articles in this topic.
Q.4 If a modifier like graphene, nanotube, gold nanoparticles and etc improved the analytical performances of the biosensors, the author should explain why that specific nanomaterial improves the analytical performances compared to other ones.
Response: In general, nanoparticles and their related entities which were employed as interface materials for sensor applications provide performance than bulk materials due to their high surface-to-volume ratio. The justification is discussed in lines numbered 96 to 105 in the revised manuscript.
Q.5 the number of the table is good enough for a review manuscript but the number of the scheme is not enough.
Response: We thank the reviewer for the suggestion. An additional figure (Figure 9) has been included in the revised version of the manuscript.
Reviewer 3 Report
I found this paper quite interesting. It is missing some reference for example on row 128 the statement "While chemical sensors are cost-effective and stable, they do not possess 128 the specificity and sensitivity of biosensors." need a reference. Similar situation form row 144 to 157 . Please add reference to all statements trough out the manuscript. Some minor english spelling like in row 283.
Author Response
Response to Reviewer-3 comments
Q.1 It is missing some reference for example on row 128 the statement "While chemical sensors are cost-effective and stable, they do not possess 128 the specificity and sensitivity of biosensors." need a reference.
Response: We thank the reviewer for pointing out the missing reference. The statement is related to the reference number i.e. 23 (row 81) and has been cited in the revised manuscript.
Q.2 Similar situation from row 144 to 157. Please add reference to all statements trough out the manuscript.
Response: The statements from rows 144 to 157 are now 96 to 109 and the appropriate reference (Reference number 31) has been cited in the revised manuscript.
Q.3 Some minor english spelling like in row 283.
Response: We thank the reviewer for pointing out the errors. The spelling errors have been corrected in the revised manuscript after thorough proof-reading.
Reviewer 4 Report
The review discusses the use of immunosensors for the electrochemical detection of HbA1c. The topic is interesting, in my opinion, and this review can be published after revision.
1) How was the bibliography chosen? have the PRISMA criteria been met? how many and which data banks were used? What search criteria were used? How many articles have been selected? how many were excluded and why? This information must be added in the text
2) in general, the bibliography underlying this review seems a bit limited.
3) Considering that there are already 140 (https://www.scopus.com/results/results.uri?sort=plf-f&src=s&st1=electrochemical+detection+of+HbA1c&nlo=&nlr=&nls=&sid=36662b67bb6751ea523f8f15c3a5d599&sot=b&sdt=cl&cluster=scosubtype%2c%22re%22%2ct&sl=39&s=ALL%28electrochemical+detection+of+HbA1c%29&origin=resultslist&zone=leftSideBar&editSaveSearch=&txGid=848db6a44e84e59a7077f6319031e904) reviews on the subject, what are the peculiarities of this new review? What are the new aspects highlighted?
5) The language must be improved
6) The final information request in the journal template can be inserted
Author Response
Response to Reviewer-4 comments
Q.1 How was the bibliography chosen? have the PRISMA criteria been met? how many and which data banks were used? What search criteria were used? How many articles have been selected? how many were excluded and why? This information must be added in the text
Response: All science citation indexed articles in Science Direct, PubMed, and google scholar on electrochemical sensors for HbA1c published from the year 2000 till date have been used for the review. Sensors fabricated with either chemical or biological sensing elements have been considered for review. Nano-interfaced and point-of-care sensors for HbA1c have been emphasized upon. Similar interfaces and electrode modification strategies have been excluded for minimizing redundancy. This information has now been included in the revised manuscript.
Q.2 In general, the bibliography underlying this review seems a bit limited.
Response: This is a concise review based on electrochemical sensors for glycated hemoglobin (HbA1c) which highlights the most important and precise literature from 2000 to date.
Q.3 Considering that there are already 140 (https://www.scopus.com/results/results.uri?sort=plff&src=s&st1=electrochemical+detection+of+HbA1c&nlo=&nlr=&nls=&sid=36662b67bb6751ea523f8f15c3a5d599&sot=b&sdt=cl&cluster=scosubtype%2c%22re%22%2ct&sl=39&s=ALL%28electrochemical+detection+of+HbA1c%29&origin=resultslist&zone=leftSideBar&editSaveSearch=&txGid=848db6a44e84e59a7077f6319031e904) reviews on the subject, what are the peculiarities of this new review? What are the new aspects highlighted?
Response: The uniqueness of this review lies in the framework that it is drafted taking into consideration past and present developments regarding electrochemical sensors for HbA1c detection. This article also summarizes commercially available point-of-care (PoC) devices for HbA1c sensors and discusses the challenges related to fabricating affordable PoC with respect to detection limit and sensitivity.
Q.4 The language must be improved.
Response: The language throughout the manuscript has been checked and revised wherever required as suggested.
Q.5 The final information request in the journal template can be inserted
Response: The revised manuscript has been formatted in the given journal template.
Round 2
Reviewer 4 Report
The authors have addressed most of the comments; they have also tried to make changes according to the reviewers' suggestions. After revisions, the quality of the manuscript has been adequately enhanced. Therefore, the manuscript could be considered for the publication